# Are the predicted known bacterial strains in a sample really present? A case study

**Minerva Ventolero[1☯], Saidi Wang[2☯], Haiyan Hu[3]\*, Xiaoman Li[1]\***

**1** Burnett School of Biomedical Science, College of Medicine, University of Central Florida, Orlando, Florida, United States of America, **2** Department of Computer Science, University of Central Florida, Orlando, Florida, United States of America, **3** Department of Computer Science, Genomics and Bioinformatics Cluster, University of Central Florida, Orlando, Florida, United States of America

☯ These authors contributed equally to this work.
\* xiaoman@mail.ucf.edu (XL); haihu@cs.ucf.edu (HH)

## Abstract

With mutations constantly accumulating in bacterial genomes, it is unclear whether the previously identified bacterial strains are really present in an extant sample. To address this question, we did a case study on the known strains of the bacterial species *S. aureus* and *S. epidermis* in 68 atopic dermatitis shotgun metagenomic samples. We evaluated the likelihood of the presence of all sixteen known strains predicted in the original study and by two popular tools in this study. We found that even with the same tool, only two known strains were predicted by the original study and this study. Moreover, none of the sixteen known strains was likely present in these 68 samples. Our study thus indicates the limitation of the known-strain-based studies, especially those on rapidly evolving bacterial species. It implies the unlikely presence of the previously identified known strains in a current environmental sample. It also called for de novo bacterial strain identification directly from shotgun metagenomic reads.

## Introduction

It is important to study bacterial strains. Bacterial genomes typically have a mutation rate of $10^{-7}$ to $10^{-10}$ substitutions per nucleotide per generation [1]. With such constant mutations, it is common that a bacterial species has multiple strains coexisting in an environmental or clinical sample. These coexisting strains from the same bacterial species often have different fitness and strength to survive and react to different environmental stimuli, which is one of the main causes of drug resistance, mixed infection, reinfection, etc. [2–6]. It is thus necessary to study bacterial strains.

Currently, bacterial strains in environmental and clinical samples are routinely studied by shotgun metagenomic sequencing, where the DNA mixture in a sample is shotgun sequenced by next-generating sequencers [7–10]. These sequenced DNA fragments are called reads. Reads are short, containing sequencing errors, and mixed with other reads from different species and strains in the sample. Because of the importance of bacterial strains, many studies have explored identifying known and reconstructing novel bacterial strains from shotgun metagenomic reads [11–13].

**Data Availability Statement:** The Chng datasets are available at NCBI BioProject PRJNA277905. All in house scripts used can be found at https://github.com/UCF-Li-Lab/known-Strains. Strains used in building databases are listed in S3 Table.

Other relevant information regarding dataset and tools used are included in the Materials and Methods.

**Funding:** This work was supported by the National Science Foundation (https://www.nsf.gov/) grants 1661414 (HH) 2120907 (HH), and 2015838 (XL). The funder plays no role in the study design, data collection and analysis, decision to publish, or preparation of the manuscript.

**Competing interests:** The authors have declared that no competing interests exist.

While the reconstruction of novel bacterial strains is still in its infancy, about a dozen studies have delved into identifying known strains from shotgun metagenomic reads [11, 14–22]. Many known strains of important pathogens are available in the form of genome sequences, strain gene contents, single nucleotide variation (SNV) or single nucleotide polymorphism (SNP) profiles, multiple-loci sequence typing sequences (MLSTs), and so on [11, 14–22]. These studies determine whether a known strain is present in a sample by checking whether its genome sequence, gene contents, SNVs, SNPs, MLSTs, k-mers, etc., are supported by the shotgun reads. They may claim the presence of novel strains if reads do not support any known strain in databases.

Note that it is logically incorrect to claim the presence of a known strain even if all of its gene contents, SNVs, SNPs, MLSTs, k-mers, etc., occur in the reads. For instance, assume this known strain has one thousand SNPs, and three unknown strains A, B, and C together also contain these one thousand SNPs. In this case, the available known-strain-based methods will claim the presence of this known strain in the sample, while it is also possible that this known strain is absent and the three unknown strains are present.

With the flawed logic behind the current known-strain-based methods and the fact that bacterial genomes endlessly accumulate mutations, it is unclear whether known strains identified previously are really present in an extant sample and whether existing tools can consistently identify known strains in the same samples [20, 21, 23]. To address these questions, we did a case study on the known strains in 68 atopic dermatitis (AD) related shotgun metagenomic samples. Chng et al. previously generated these samples and applied the tool Pathscope2 [16] to these samples to identify known bacterial strains [24]. They discovered five *S. aureus* strains and six *S. epidermidis* strains in these samples and studied the association of the abundance shift of these strains with AD susceptibility. We reran Pathscope2 on these samples and could identify only two of the eleven known strains reported by Chng et al. We also applied another popular tool, StrainEST [15], to these samples and could find only three of the eleven known strains (Material and Methods). Pathscope2 and StrainEST are two of the most widely cited tools for known strain identification and have been updated since their publication. We further demonstrated that none of the sixteen known strains from these three analyses (Chng et al., the rerun of Pathscope2, the application of StrainEST) was likely to occur in the samples. Our study suggests that these known strains never exist in any of the 68 current samples for the two bacterial species in this case. Despite the limited scope of this case study with only one dataset, two species and two popular tools, our study also indicates the limitation of the known-strain-based tools and the necessity to develop novel bacterial strain discovery tools for analyzing shotgun metagenomic reads.

## Results

### Reanalysis of the AD samples cannot recover most of the originally identified known strains

To study what could predispose the AD skin to flare conditions, Chng et al. studied the interflare microbiota of dormant AD skin of volunteers susceptible to AD. There were nineteen samples from AD patients and fifteen samples from healthy controls. Chng et al. did whole shotgun sequencing on these samples with two replicates for each sample and generated 589 million 100 base pairs long paired-end reads in 68 samples. They applied Pathoscope2 to study bacterial strain differences in AD and healthy control samples and identified five *S. aureus* strains and six *S. epidermidis* strains. However, since the *S. aureus* MSHR11 strain was re-classified as *S. argenteus* and the *S. epidermidis* strain genomes of A487 and M23864_W1 could

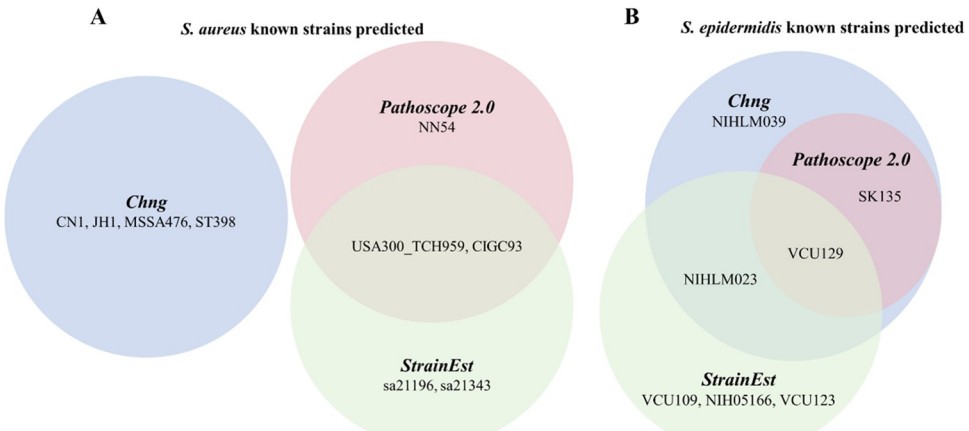

**Fig 1. Predicted known strain in three analyses.** (A) *S. aureus* known strains predicted and (B) *S. epidermidis* known strains predicted by Chng et al., the rerunning of PathoScope 2.0 and StrainEst.

not be downloaded, only four of the original reported *S. aureus* strains and four of the original reported *S. epidermidis* strains were included in this study (Fig 1A and 1B).

To investigate whether we could rediscover the same strains in the same samples, we reanalyzed these datasets with the same tool, Pathoscope2. Since the gi_taxid_nucl.dmp.gz file from National Center for Biotechnology Information (NCBI) was no longer available, the created Pathoscope2 MySQL library could not be downloaded, and the library building utilizing these resources in Pathoscope2 could not be carried out. Instead, we used an alternate database of the strain genomes with taxon IDs prepended already (Materials and Methods). The reanalysis by Pathoscope2 did not identify any of the four claimed *S. aureus* strains (Fig 1A). For *S. epidermidis*, the reanalysis identified two and only two of the four reported strains (Fig 1B). The discrepancy in the identified strains was unlikely influenced by the used database since we compared only the strains shared by the original database and the database we used.

Attempting to resolve the difference between the predictions from the original study and the predictions from the reanalysis with Pathoscope2, we applied a different tool, StrainEst [15], to the same samples. Pathoscope2 determines whether a known strain occurs in a sample by mapping reads to all known strain genomes in a database and determining which strains may be present based on the mapped reads and their mapping scores to each strain in the database. On the other hand, StrainEst performs strain analysis based on the SNV matrices inferred from known strain genomes in an input database. This input database consists of a set of strain genomes and a designated species representative. Only strain genomes that pass the criteria that their distance to the species representative genome was smaller than a cutoff would be used for the SNV profiling. This cutoff means that while the original list of strain reference genomes was similar to that used in the Pathoscope2 reanalysis, the StrainEst database in the SNV profiling and read mapping steps was essentially a filtered list of known strains. StrainEst did not predict any of the four *S. aureus* strains from Chng et al., while it predicted two *S. aureus* strains from the reanalysis of Pathoscope2 (Fig 1A). For *S. epidermidis*, StrainEst identified one strain shared by Chng et al. and the Pathoscope2 reanalysis, and one strain shared with Chng et al. only, together with three additional strains not reported above (Fig 1B). The shared strains by the reanalysis of Pathoscope2 and StrainEst suggested that these strains may be involved in AD. At the same time, the difference in the predictions from the two tools indicated that the algorithms behind the known strain analysis might often generate false positive predictions.

The above three approaches generated quite different predictions (Fig 1). Among the predicted nine *S. aureus* strains and seven *S. epidermis* strains, only 31.3% of the strains were reported by at least two analyses. The percentage of the shared *S. aureus* strains was lower than that of the shared *S. epidermis* strains, possibly due to their different numbers of known strains and genome sizes. We used 214 known *S. aureus* strains and 61 *S. epidermidis* known strains, and the *S. aureus* genome is about 15% longer than the *S. epidermidis* genome. More strains and longer genomes might confuse the tools and contribute to the discrepancy of the identified known strains. The predicted strains were more consistent between the reanalysis by Pathoscope2 and StrainEst, likely due to the same input database of known strains.

## Reads in the pooled sample do not support any of the identified known strains

Although most strains are present in only one analysis, four are identified in two analyses, and one is reported by all three analyses. Are these shared strains more likely to be present in the samples? To address this question, we studied all nine *S. aureus* strains and all seven *S. epidermis* strains (Fig 1) in the pooled sample, which combines the shotgun reads from all 68 samples as one sample. We found that the reads in the pooled sample supported none of the above strains.

We hypothesized that if a known strain occurred in these 68 samples, the majority of its unique SNPs would have similar coverage in the pooled sample (Material and Methods). In other words, the coverage of these SNPs unique to a strain should be present in a similar number of reads in the pooled sample, which means the number of reads containing the unique SNPs of a strain statistically follows a uniform distribution (Table 1 and S1 Table). Note that because we do not know all strains of a species, an unknown strain may contain the unique SNPs defined with all known strains of a species. We found that for all strains reported in the above three analyses, their unique SNPs did not follow a uniform distribution based on the Kolmogorov–Smirnov test. Almost all strains had a p-value smaller than 2E-17, indicating the non-uniform coverage of the unique SNPs of every strain and the absence of these known strains. The only two strains having a large p-value were JH1 and NN54. However, the large p-value was because 98.5% and 99.6% of their unique SNPs had zero coverage. In other words, almost all unique SNPs of these two strains did not occur in the mapped reads, indicating that JH1 and NN54 were also absent. For both species, the p-value was smaller for strains with

**Table 1. The unique SNPs in the predicted *S. aureus* known strains.**

| *S. aureus* known strains | | Uniform Distribution (p-value) | #original SNPs | #unique SNPs | #unique SNPs present | %unique SNPs present | #unique SNPs absent | %unique SNPs absent | Average coverage |
|---|---|---|---|---|---|---|---|---|---|
| Chng | CN1 | 9.7E-34 | 21016 | 3210 | 70 | 2.18 | 3140 | 97.8 | 36.4 |
| | JH1 | 0.03 | 24308 | 882 | 13 | 1.47 | 869 | 98.5 | 13.8 |
| | MSSA476 | 1.0E-17 | 20886 | 3321 | 40 | 1.20 | 3281 | 98.8 | 25.5 |
| | ST398 | 4.9E-156 | 54316 | 35753 | 265 | 0.74 | 35488 | 99.3 | 41 |
| Pathoscope2 | CIGC93 | 4.7E-86 | 19194 | 3055 | 100 | 3.27 | 2955 | 96.7 | 17.2 |
| | NN54 | 0.01 | 23597 | 889 | 4 | 0.45 | 885 | 99.6 | 15.6 |
| | USA300_TCH959 | 7.1E-100 | 20802 | 3515 | 153 | 4.35 | 3362 | 95.6 | 19.1 |
| StrainEst | sa21196 | 6.0E-27 | 20890 | 3847 | 65 | 1.69 | 3782 | 98.3 | 23.3 |
| | sa21343 | 1.6E-22 | 22939 | 4483 | 37 | 0.83 | 4446 | 99.2 | 32.4 |
| | CIGC93 | 4.7E-86 | 19194 | 3055 | 100 | 3.27 | 2955 | 96.7 | 17.2 |
| | USA300_TCH959 | 7.1E-100 | 20802 | 3515 | 153 | 4.35 | 3362 | 95.6 | 19.1 |

more unique SNPs that occurred in the reads than strains with fewer unique SNPs that were present in the reads. Interestingly, the strain VCU129 had the most unique SNPs with non-zero coverage and was the only strain predicted by all three analyses. It was also evident that the known strains predicted by StrainEst had more unique SNPs with non-zero coverage than the known strains from PathoScope2, suggesting that StrainEst focuses more on individual SNPs to claim the presence of a known strain. Consistently, the four known strains predicted by StrainEst and PathoScope2 also had relatively larger numbers of unique SNPs with non-zero coverage.

Although shotgun reads theoretically should uniformly cover a genome, they might not do so in practice because of the genome heterogeneity [25–27]. We thus relaxed the uniform assumption and hypothesized that a large fraction of unique SNPs in a known strain should be present in the shotgun reads if this strain was present in the 68 samples and had a moderate coverage. We found that this was not the case for all reported nine *S. aureus* known strains in the sample. At least 95.6% of unique SNPs in each S. aureus known strain were absent in the mapped reads, suggesting that these known strains may be absent in the sample (S1 Table). For instance, for the reported JH1 strain, 98.5% of its unique SNPs were not observed in reads, while the average coverage of its present unique SNPs was 13.8X. Based on the Poisson distribution assumption of the coverage of an SNP, the chance that we could not observe a unique SNP of the JH1 strain in reads is 1.0E-6. The p-value that 98.5% of the 882 unique SNPs in JH1 were not observed in reads was thus 0 based on the binomial testing.

We had a similar observation of a large number of absent unique SNPs in reads for the seven *S. epidermidis* known strains (S1 Table). Compared with *S. aureus*, *S. epidermidis* strains had a smaller percentage of absent unique SNPs in reads. However, 23.5% to 92.7% of unique SNPs were still absent in reads. Given the percentage of the absent unique SNPs and the corresponding strain coverage based on its present unique SNPs, the p-value to observe such a large portion of absent unique SNPs in each known strain was still 0, suggesting that these S. epidermidis known strains may not have been present in these 68 samples.

Overall, although different analyses reported known strains of *S. aureus* and *S. epidermis* in these 68 samples, we could not find any support for their existence in terms of uniform coverage of their unique SNPs in the pooled sample. Moreover, too many unique SNPs in each strain had zero coverage in reads (i.e., absent in reads) even when the strain had moderate coverage, supporting that no reported known strain was present in these samples.

## Reads across individual samples do not support the presence of any identified known strain

We assessed whether the known strains identified above were present by studying their coverage in the 68 individual samples. Different from the pooled sample, individual samples provide information about how the strain abundance varies across samples. If a known strain is present in a sample, all its unique SNPs are present in this sample, although a unique SNP may have zero coverage. Therefore, the coverage of each unique SNP of a strain is expected to fluctuate consistently with the coverage of this strain across samples. In other words, although the coverage of these unique SNPs in a strain may not be uniformly distributed, the coverage of every pair of its unique SNPs is expected to be highly correlated across samples. This high correlation is because the coverage of every unique SNP of a strain represents the coverage of this strain, and the coverage vector of every unique SNP of a strain thus represents the same strain coverage variation across samples. The coverage vector of an SNP refers to the vector composed of the number of reads mapped to this SNP position and containing this SNP across samples. We considered only the present unique SNPs (unique SNPs with non-zero pooled coverage) in

**Table 2.  The correlated unique SNP pairs in the predicted _S. aureus_ known strains.**

| _S. aureus_ Known strains | Chng | | | | Pathoscope2 | | | StrainEst | | | |
|---|---|---|---|---|---|---|---|---|---|---|---|
| | CN1 | JH1 | MSSA476 | ST398 | CIGC93 | NN54 | USA300_TCH959 | sa21196 | sa21343 | CIGC93 | USA300_TCH959 |
| #Pairs | 2415 | 78 | 780 | 34980 | 4950 | 6 | 11628 | 2080 | 666 | 4950 | 11628 |
| #significant pairs | 391 | 25 | 114 | 3914 | 1037 | 4 | 9408 | 392 | 169 | 1037 | 9408 |
| %significant pairs | 16.2 | 32.1 | 14.6 | 11.2 | 21.0 | 66.7 | 80.9 | 18.9 | 25.4 | 21.0 | 80.9 |

every known strain here, as no data was available to measure the coverage of the absent unique SNPs in these strains.

We found that the majority of unique SNP pairs did not have correlated coverage vectors for seven of the nine _S. aureus_ known strains and six of the seven _S. epidermidis_ known strains (Table 2 and S2 Table). No more than 32.1% of unique SNP pairs had a significant correlation (False discovery rate <0.01) for the seven _S. aureus_ known strains. In comparison, this number was 19.8% for the six _S. epidermidis_ known strains, suggesting that these known strains were likely to be absent. Two _S. aureus_ strains, NN54 and USA300_TCH959, had 66.7 and 80.9% unique SNP pairs with significant correlation while having only 4 and 153 unique SNPs in the corresponding strains, respectively. Because more than 95.6% of their unique SNPs were absent, the correlated unique SNP pairs accounted for too few possible pairs in these two strains to claim their presence (Table 2). One _S. epidermidis_ strain, NIHLM023, did have 44.7% of unique SNP pairs correlated, and 58.6% of its unique SNPs were observed in the reads. Although NIHLM023 had a much larger proportion of its unique SNPs in the reads and a larger fraction of the present unique SNP pairs had a good correlation (False discovery rate <0.01), there were still more than 55.3% of unique SNP pairs not correlated well. NIHLM023 was thus unlikely to be present.

We also studied whether the known strains identified in multiple analyses were more reliable than those identified in only one analysis. We found that the strains identified in at least two analyses did not necessarily have a higher percentage of unique SNP pairs with a correlated coverage across samples. For instance, although VCU129 was identified in all three analyses, it had the second smallest percentages in the Chng's and StrainEst analyses while the smallest percentage in the reanalysis with PathoScope2. Another example was the strain CIGC93, which was identified in two analyses but had a smaller percentage than other strains identified in both analyses. In general, the strains identified in at least two analyses had either a larger percentage of unique SNPs present in the reads and/or a higher percentage of correlated unique SNP pairs. In this sense, the strain ST398 might be an interesting case, which had the highest number of SNPs and unique SNPs present in reads but the lowest percentage of unique SNPs present in reads in Chng's analysis and was not predicted by any other analysis. Since StrainEst tended to predict strains with more unique SNPs present in reads, we checked whether StrainEst filtered the strain ST398 or it included this strain in its input database. We found that the ST398 was in the filtered database input to StrainEst, which suggested that it is not the number of unique SNPs in reads but the percentage of unique SNPs present in reads that contributed to the predictions of StrainEst.

## Discussion

We hypothesized that predicted known bacterial strains might be absent in a new clinical or environmental sample, especially for the fast-evolving pathogens. We tested this hypothesis with 68 AD samples as a case study. We showed that we could not find most of the reported known strains in the original study, even with the same tool on the same datasets. We also

found that all identified known strains from the original study and those from the reanalysis of the same data with two tools were likely to be absent in these samples. Our study thus demonstrated the limitation of the known-strain-based strategy to interrogate the strain diversity in microbiomes and the great necessity to identify novel strains directly from shotgun reads.

It is not surprising that the reported known strains in certain samples are absent in these samples. The existing known-strain-based tools often examine the mapped reads to determine whether a known strain is present. Because of sequencing errors in reads, with a high sequencing coverage of a species, the shotgun reads are likely to support every unique region in a strain. However, the presence of every unique region of a strain in reads will not prove the existence of such a known strain in the sample unless a long sequence consisting of each of the unique regions in this strain is available from long reads or the assembled genomes, as illustrated in the example in the introduction section. With such long sequences, current known-strain-based tools might likely to rule out the absent strains with all of their unique regions present in reads. To make the analysis of known strains even more difficult is the limited number of known strains available and the constantly mutated bacterial genomes. Such difficulties will result in mistaking an unknown strain for a close known strain in the database or misinterpreting the shared regions by an unknown strain as the unique regions in the known strain.

Our study also indicates the potential biases in a known-strain-based strategy. Such a strategy already has candidate known strains in mind before it attempts to justify their existence in shotgun reads. In other words, such a strategy is misleading because the strains of a bacterial species are so similar to each other that many reads can be mapped to multiple strains. Moreover, even if a read perfectly maps to one unique strain, it does not guarantee that this read is from this strain, as there are unknown strains to which this read may be perfectly mapped. In addition, such a strategy never reconstructs the complete strain genomes when trying to identify known strains, which leaves the presence of known strains as, at most, a best guess instead of real proof.

It may thus be more meaningful to de novo identify novel strains instead of known strains directly from shotgun metagenomic reads. About a dozen tools are available for novel strain identification [11–13]. We recommend using the recently developed tools with good accuracy and applicability to multiple samples [28, 29]. Note that the development of the novel strain identification methods is still in its infancy, and new tools may be developed with better performance in the future. In addition, with improved long-read sequencing and decreased sequencing cost, we could directly sequence the entire novel strain genomes, which may eventually address the strain identification problem in shotgun metagenomic sequencing samples.

## Material and methods

### The AD samples

We obtained the 68 AD samples from Chng et al. [24], available at NCBI BioProject PRJNA277905 (https://www.ncbi.nlm.nih.gov/bioproject/277905). There are 19 samples from AD patients and 15 samples from healthy controls. For each sample, there were two replicates shotgun sequenced. We considered each replicate as a different sample in this study. We downloaded the raw read data from the NCBI BioProject and applied Trimmomatic [30] to clean the reads. The command used to clean the reads by Trimmomatic was:

java -jar <path to trimmomatic.jar> PE -phred33 <pooled raw pair1.fastq> <pooled raw pair2.fastq> <paired output1> <unpaired output 1> <paired output 2> <unpaired output 2> ILLUMINACLIP:<location of TruSeq3-PE.fa>:2:30:10 LEADING:3 TRAILING:3 SLIDINGWINDOW:4:20 MINLEN:36.

## Strain analysis

We predicted strains in these 68 AD samples using the tools Pathoscope2 and StrainEst [31] and the cleaned reads by Trimmomatic [30]. We chose Pathscope2 and StrainEST for known strain prediction because they were the most highly cited known-strain-based tools that could predict the presence of known strains on the genome scale (not locally) and were kept updated after their publications. Pathoscope2 needs target libraries to be input. We could directly use either the NCBI file or a previously created Pathocope2 MySQL database as the target libraries. However, creating the libraries through either method was impossible because the NCBI file gi_taxid_nucl.dmp.gz and the Pathoscope MySQL database (ftp://pathoscope.bumc.bu.edu/data/pathodb.sql.gz) were not available to download. Access to this NCBI file and MySQL database is needed to append taxIDs, which is a crucial step in creating the libraries. The Pathoscope2 simplified database version with the tax IDs already prepended (ftp://pathoscope.bumc.bu.edu/data/nt_ti.fa.gz) was also inaccessible. To address these issues, an alternative library database was created, similar to the simplified version. To do this, We collected the listed strains in the study of Byrd et al. [32], compiling the genome files in a directory for each species with the NCBI refseq genomes. These collected strain genomes comprised the genomes for 214 *S. aureus* known strains and 61 *S. epidermidis* known strains, which included all known strains reported by Chng et al. except for the *S. aureus* strain MSHR11 that was re-classified as *S. argenteus* and the *S. epidermidis* strains A487 and M23864_W1 that could not be found in the NCBI RefSeq assembly_summary file (S3 Table). Using the pasteTaxID script (https://github.com/microgenomics/pasteTaxID), tax IDs were prepended to the strains using the command: bash pasteTaxID.bash–workdir [directory_fastas]. All strains with their tax IDs were then compiled into one fasta file labeled as AurEpix_ti.fasta to be used as the genomeFile for the library building in Pathoscope2. To create the target libraries, the Pathoscope2 command used was: pathoscope LIB -genomeFile <directory of the AurEpix_ti.fasta> -taxonIds <list the tax IDs of the strains to be included in the library separated by comma> -outPrefix <prefix to use for the library being created> -outDir <output directory>.

We followed Chng et al.'s approach for strain analysis, where reads were mapped to the *S. aureus* target and used the *S. epidermidis* library as a decoy and vice versa. For the steps Patho-Map and PathoID, we followed the instructions indicated in their tutorial wiki (https://github.com/PathoScope/PathoScope/wiki). From the resulting PathoID tsv file, only strains with ≥ 0.05 percentage of read alignment (final guess column) were considered the main strains identified in this re-analysis. Note that we did not screen for contaminated human reads as Chng et al. before applying Pathoscope2 because Chng et al. were trying to narrow down the species and strains of interest while we already had the strains of interest. Moreover, the strain genomes of the two species we considered share low similarity with the human genome (with only one segment in one of the *S. epidermidis* strain genomes and none in any *S. aureus* strain genome having a blast e-value smaller than 1 when compared with the human genome) and the contaminated human reads would unlikely be mapped to these collected strain genomes.

For StrainEst (the Docker version), all analyses followed the commands in the StrainEst [31] paper, starting from the selection of strain genomes to be used for the SNV profiling. We first installed Mash (https://github.com/marbl/Mash/releases/tag/v2.3). We then computed the pairwise Mash distances of strains, where strains with a distance ≤0.1 from species representative (SR) were kept for the SNV profiling step. The strain genomes used for distance analysis were the same list of strains from Byrd et al. [32], and the SR was NC_007795.1 (https://www.ncbi.nlm.nih.gov/nuccore/NC_007795.1; RefSeq: https://ftp.ncbi.nlm.nih.gov/genomes/all/GCF/000/013/425/GCF_000013425.1_ASM1342v1/) and NZ_CP035288 (https://www.

ncbi.nlm.nih.gov/nuccore/NZ_CP035288.1; RefSeq: https://ftp.ncbi.nlm.nih.gov/genomes/all/GCF/006/094/375/GCF_006094375.1_ASM609437v1/) for *S. aureus* and *S. epidermidis*, respectively. Here 119 *S. aureus* and 42 *S. epidermidis* strains were used at the SNV profiling step. The reported strains by Chng et al. were included purposefully in the filtered database to ensure they were part of the read mapping. The StrainEst pipeline was followed starting from the creation of the MR.fasta for the SNV profiling step to the output of strains and their abundances using default parameters. For the metagenome alignment step, the chosen strain genomes were based on the clustering output from the SNV profiling step, where two from each cluster with the lowest distances were chosen to be used for the MA.fasta generation. Only strains with $\geq 0.05$ abundances in the final StrainEst output were considered as the main strains identified here.

## Coverage in the pooled sample and individual samples

Coverages of the known strains in the dataset were determined by first obtaining the strain SNPs by aligning each known strain genome with their species reference genomes. We used NC_007795.1 and NZ_CP035288 as the species reference genome for *S. aureus* and *S. epidermidis*, respectively. We then defined unique SNPs in each of the nine *S. aureus* and each of the seven *S. epidermidis* strains by requiring the SNPs to only occur in the strain under consideration. We considered only the unique SNPs occurring in the positions covered by the species reference genome. Next, we mapped the cleaned reads in the pooled samples or an individual sample to the species reference genome and counted the nucleotide frequencies of the four types of nucleotides. Finally, we considered the corresponding nucleotide frequency at the corresponding genome position as the coverage of a unique SNP in a known strain. For instance, for a unique SNP 'C' at position 100 of the *S. aureus* reference genome, the frequency of 'C' at this position based on the mapped reads was the coverage of this SNP. In this way, unique SNPs with a coverage 0 meant that they did not occur in any read. Coverage and SNP frequency analysis were done using in-house scripts at https://github.com/UCF-Li-Lab/known-Strains.

## Statistical analysis

We studied whether the coverage of the unique SNPs in a known strain is uniformly disturbed. The uniform distribution was determined by the Kolmogorov-Smirnov test. To see whether the coverage of unique SNPs of a strain was correlated, we calculated the Pearson's correlation coefficient of every pair of unique SNPs that were present in reads in every strain across the 68 samples. The significance of the correlation was determined using the formula $1 - pt\left(r * \sqrt{\frac{66}{1 - r * r}}, 66\right)$ in python, where *r* referred to the calculated correlation. With a large number of unique SNPs in a known strain, we selected the significant correlations for each strain based on the Benjamini and Hochberg [33] method to control the false discovery rate at 0.001.

## Supporting information

**S1 Table. Known strain unique SNP coverages.**
(DOCX)

**S2 Table. Known strain correlations.**
(DOCX)

**S3 Table. Strains used in database.**
(DOCX)

## Author Contributions

**Conceptualization:** Haiyan Hu, Xiaoman Li.

**Data curation:** Minerva Ventolero, Saidi Wang.

**Formal analysis:** Minerva Ventolero, Saidi Wang.

**Funding acquisition:** Haiyan Hu, Xiaoman Li.

**Methodology:** Haiyan Hu, Xiaoman Li.

**Supervision:** Haiyan Hu.

**Validation:** Minerva Ventolero, Saidi Wang.

**Visualization:** Minerva Ventolero, Saidi Wang.

**Writing – original draft:** Minerva Ventolero, Haiyan Hu.

**Writing – review & editing:** Minerva Ventolero, Saidi Wang, Haiyan Hu, Xiaoman Li.

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
