## [Decision Letter · Decision Letter 0]

16 Jan 2023

PONE-D-22-31800Are known bacterial strains really present?PLOS ONE

Dear Dr. Li,

Thank you for submitting your manuscript to PLOS ONE. After careful consideration, we feel that it has merit but does not fully meet PLOS ONE’s publication criteria as it currently stands. Therefore, we invite you to submit a revised version of the manuscript that addresses the points raised during the review process.

In addition to the reviewers concerns, further information about current de novo bacterial strain identification directly from shotgun metagenomic reads and what approaches can be taken to do this should be added. The discussion section is very short and will benefit from a discussion of how current de novo approaches can help alleviate the problem addressed in the paper. Please make sure you answer all the concerns for further consideration of the paper.

We look forward to receiving your revised manuscript.

Kind regards,

Achraf El Allali, PhD

Academic Editor

PLOS ONE

Journal Requirements:

Additional Editor Comments:

In addition to the reviewers concerns, I would ask that you give further information about current de novo bacterial strain identification directly from shotgun metagenomic reads and what approaches can be taken to do this. The discussion section is very short and will benefit from a discussion of how current de novo approaches can help alleviate the problem addressed in the paper. Please make sure you answer all the concerns for further consideration of the paper.

Reviewers' comments:

Reviewer's Responses to Questions

**Comments to the Author**

1. Is the manuscript technically sound, and do the data support the conclusions?

Reviewer #1: Partly

Reviewer #2: Partly

2. Has the statistical analysis been performed appropriately and rigorously? 

Reviewer #1: N/A

Reviewer #2: N/A

3. Have the authors made all data underlying the findings in their manuscript fully available?

Reviewer #1: No

Reviewer #2: Yes

4. Is the manuscript presented in an intelligible fashion and written in standard English?

Reviewer #1: No

Reviewer #2: Yes

5. Review Comments to the Author

Reviewer #1: The manuscript “Are known bacterial strains really present?” is good but this reviewer has some major concerns. We know the facts that the bacterial genomes are higly mutated; In bacterial genomes there are certain regions which are variable and some are constant so variability arisis and henece different strains; The analysis depends upon the use of databases used for either alignment or annotation for identification. Probably the authors should try to use different databases; why these comments?- as I am not having data analysis files. so unable to comment. If you have them please check for - how much variability is there from the original analysis; What is being used (Databases) initially for analysis. Keep the fact that the data, databases and information is growing and available day by day, so some variability is expected but within limits; The tools Pathoscope and IsoRead are used for the analysis, so right use of databases is very important for any type of analysis; compare the discussion with (https://doi.org/10.3389/fcell.2021.736994) (DOI: 10.1111/jam.15469) (https://doi.org/10.1007/s11356-022-22197-4)

Reviewer #2: This is an interesting study and the authors have suggested an interesting idea. The author emphasizes the drawbacks of utilizing known strains for studying bacterial genomes in shotgun metagenomic samples, as bacterial genomes are continually evolving, making it uncertain whether previously identified known strains can provide meaningful insights for understanding current samples. The paper is generally well structured. However, in my opinion the paper has some shortcomings in regard to the analyses :

1- Can you provide an explanation for why the preprocessing parameters in the original paper differ from those used by the authors?

2- It's not clear if the data used was screened for human contamination, which could affect the results. If it was, please include this information in the methods section.

3- I recommend revising the methods section, specifically the section on strain analysis.

6. PLOS authors have the option to publish the peer review history of their article (what does this mean?). If published, this will include your full peer review and any attached files.

Reviewer #1: No

Reviewer #2: **Yes: **Laamarti Mariam

---

## [Author Response · Author response to Decision Letter 0]

8 Mar 2023

Title: Are known bacterial strains really present?

Authors: Minerva Ventolero, Saidi Wang, Haiyan Hu, Xiaoman Li

Dear Dr. Achraf El Allali, 

Thank you so much for the opportunity to revise the manuscript! We appreciate the constructive comments and suggestions from the reviewers. We have improved our manuscript to clarify the points which were not described clearly or properly, and responded to the points raised by the reviewers below and addressed them in the manuscript. We hope the modified manuscript can be accepted. 

Best Regards,

Xiaoman Li

Additional Editor Comments:

In addition to the reviewer’s concerns, I would ask that you give further information about current de novo bacterial strain identification directly from shotgun metagenomic reads and what approaches can be taken to do this. The discussion section is very short and will benefit from a discussion of how current de novo approaches can help alleviate the problem addressed in the paper. Please make sure you answer all the concerns for further consideration of the paper.

Response: Thank you for the suggestion. We have edited the manuscript to include a brief discussion on de novo approaches for strain identification. 

Reviewer 1 

Comment 1: The manuscript “Are known bacterial strains really present?” is good but this reviewer has some major concerns. We know the facts that the bacterial genomes are higly mutated; In bacterial genomes there are certain regions which are variable and some are constant so variability arisis and henece different strains; The analysis depends upon the use of databases used for either alignment or annotation for identification. Probably the authors should try to use different databases; why these comments?- as I am not having data analysis files. so unable to comment. If you have them please check for - how much variability is there from the original analysis; What is being used (Databases) initially for analysis. Keep the fact that the data, databases and information is growing and available day by day, so some variability is expected but within limits; The tools Pathoscope and IsoRead are used for the analysis, so right use of databases is very important for any type of analysis; compare the discussion with (https://doi.org/10.3389/fcell.2021.736994) (DOI: 10.1111/jam.15469) (https://doi.org/10.1007/s11356-022-22197-4)

Response: Thanks for the great comments. Yes, you are right that the analysis of known strains depends on the quality of the known strain genomes (the database) used. The annotated genome sequences stored at NCBI are often considered as gold standard (for instance, Chng et al. used the genomes from NCBI as well), the quality of which is more reliable than the genome sequences stored elsewhere. This is why we choose the annotated genome sequences at NCBI instead of those at elsewhere for the known strain analysis. Moreover, because we wanted to recreate the analysis of the previous study by Chng et al. 2016, which also used the annotated strain genomes at NCBI, we tried our best to replicate the set-up as much as possible, making sure that all specifically identified strains from the analysis they performed and reported that were still available in NCBI GenBank were part of the list of strains included in the recreated list of the genomes.

About the variability issue, these genomes we used are the reference genomes stored at NCBI, which normally do not change across time. If you are talking about the difference between two strain genomes of the two species, there are about 0.8% to 2% positions are different between a S. aureus strain with the species reference genome, and 0.3% to 2.6% positions are different between a S. epidermidis strain with the species reference genome. In terms of the difference between the list of strain genomes we used and those Chng et al. used, our list of the strain genomes for the two species targeted included every reported strains by Chng et al. and more genomes than Chng et al.’s list. The strain genomes we used were all from the latest curated list of strain genomes by NCBI RefSeq. 

Thanks for the several interesting papers you mentioned. We are not sure how the diversity issue discussed in these paper relate to what we considered here. We focused on two species and their strains in the shotgun metagenomic reads, and did not focus on the general composition of the metagenomic samples in this study. Again, we completely agree that the genome sequences used affect the analysis results. We thus used the last version of the more reliable NCBI refseq genomes. Genomes from other resources can be explored, which may be problematic here as we are not so sure about their qualities. 

Reviewer 2: 

This is an interesting study and the authors have suggested an interesting idea. The author emphasizes the drawbacks of utilizing known strains for studying bacterial genomes in shotgun metagenomic samples, as bacterial genomes are continually evolving, making it uncertain whether previously identified known strains can provide meaningful insights for understanding current samples. The paper is generally well structured. However, in my opinion the paper has some shortcomings in regard to the analyses :

1- Can you provide an explanation for why the preprocessing parameters in the original paper differ from those used by the authors?

Response: Thank you for the comment. The parameters are the same except the input strain genomes are not completely the same as those in the Chng et al’s study, because the old genome file used by the Pathoscope2 tool in Chng et al.’s study is not supported by NCBI anymore. In brief, we tried to recreate as best we could the analysis originally performed by Chng et al., 2016. Due to some challenges that were encountered when using the tool Pathoscope 2 that we have explained in detail in the methods Section, a slight modification was performed. The main problem was the file that Pathoscope 2 was programmed to use from NCBI and which was discontinued and no longer made available by NCBI, and so we used an alternative approach to create the database list but making sure that the genomes we used were still form NCBI like the Chng et al.’s study. We also made sure that all available genomes of strains reported by Chng et al. were included in the database we created and used for the analysis. The tool StrainEst was not used in the original study but was included in our study as another tool of similar approach for strain analysis identification for comparative purposes only in our study.

2- It's not clear if the data used was screened for human contamination, which could affect the results. If it was, please include this information in the methods section.

Response: Thank you for the insightful comment. For the data screening for human contamination, it is often for the purpose of understanding the general composition of a metagenomic sample. In this study, our goal is to see which known strain of the two species is present. We no longer performed the screening of human information as the tools are designed to map sequences to a specific set of strain genomes that are the target of the analysis. A human DNA fragment will in general not be mapped to a strain genome here. In fact, we blasted each of these strain genomes we used against the human hg19 genome and found one and only one segment of 30 base pairs long with the e-value smaller than 1.0 was found in one of the S. epidermis strain genomes. More importantly, to claim the presence of a strain depends on the occurrence of many of the DNA fragments of this strain in the reads, which will not be affected by these contaminant (otherwise, a strain genome of these two species have a large overlap with the human genome). We have provided the information accordingly. 

3- I recommend revising the methods section, specifically the section on strain analysis.

Response: Thank you for the suggestion. We have edited the methods section on strain analysis according to suggested changes.

---

## [Decision Letter · Decision Letter 1]

12 Jul 2023

PONE-D-22-31800R1Are known bacterial strains really present?PLOS ONE

Dear Dr. Li,

Thank you for submitting your manuscript to PLOS ONE. After careful consideration, we feel that it has merit but does not fully meet PLOS ONE’s publication criteria as it currently stands. Therefore, we invite you to submit a revised version of the manuscript that addresses the points raised during the review process. I apologize for the delay, but i had a hard time securing reviewers in the field. Now that i have received the reviews, please try to send your reponse as soon as possible. To avoid any further delay, please check your paper using grammar software or professional English proofreading service to avoid further delay before your next submission.

We look forward to receiving your revised manuscript.

Kind regards,

Achraf El Allali, PhD

Academic Editor

PLOS ONE

Reviewers' comments:

Reviewer's Responses to Questions

**Comments to the Author**

1. If the authors have adequately addressed your comments raised in a previous round of review and you feel that this manuscript is now acceptable for publication, you may indicate that here to bypass the “Comments to the Author” section, enter your conflict of interest statement in the “Confidential to Editor” section, and submit your "Accept" recommendation.

Reviewer #1: (No Response)

Reviewer #3: (No Response)

2. Is the manuscript technically sound, and do the data support the conclusions?

Reviewer #1: Partly

Reviewer #3: Partly

3. Has the statistical analysis been performed appropriately and rigorously? 

Reviewer #1: Yes

Reviewer #3: Yes

4. Have the authors made all data underlying the findings in their manuscript fully available?

Reviewer #1: (No Response)

Reviewer #3: Yes

5. Is the manuscript presented in an intelligible fashion and written in standard English?

Reviewer #1: No

Reviewer #3: Yes

6. Review Comments to the Author

Reviewer #1: The article entitled “Are known bacterial strains really present?” seems to be interesting for the research scientific community. In the revised current form authors have shown the good potential in improving this article, but still, it is in lack of structured. This reviewer thinks this article can be improve more by summarizing and pointing few concerns. Start from the abstracts, the provided abstracts could be further improved by incorporating the following elements: (1) The abstract should start with a clear and concise objective statement that highlights the specific research question or problem being addressed. This will help readers understand the purpose of the study right from the beginning. (2) Provide a brief description of the methods or approaches used in the study in one or two statements. This should include information on the data collection process, analysis techniques, and any specific tools used. (3) Instead of just stating that known strains cannot be predicted and are different across tools, provide more specific and quantitative information about the results obtained. This will provide the readers with a clearer understanding of the findings. (4) Discuss the broader implications of the study findings in a single statement. (5) Conclude the abstract by highlighting potential future directions or recommendations based on the study's outcomes. (6) In introduction, there is an excessive use of citations/references. In example, only first paragraph of the introduction including 14 references in 7 lines. Please use the citation/references according to the journal guidelines and where appropriate. (7) In introduction where stated “We reran Pathscope2 on these samples …” please clearly mention the validity of findings. Adding potential reasons for the discrepancies and significance of findings. This reviewer suggests to revise thoroughly the introduction section by mentioning the Objective statement with limitations and significances at the end of introduction; (8) Furthermore, why it is so important to use the data from study of Chng et. al, published in 2016. Please consider using some stats and data comparison from the recent scientific study from last 5 years. (9) This reviewer suggests to revise your references very carefully and according to the journal style. Most of the references used in this study are out dated. Please defend your research innovation by comparing the study from the reference list of last five years research. (10) Please revise the manuscript carefully, and avoid the grammar mistakes. I suggest you to invite a native speaker to proofread this article before further submission, or use any official English language proofread services.

Best of Luck!

Reviewer #3: The case study described aimed to answer the title question by revisiting the known strains of S. aureus and S. epidermidis in atopic dermatitis shotgun metagenomic samples. The findings of the study revealed that even using the same analysis tools, the previously reported known strains could not be predicted consistently in the same samples. Additionally, different tools gave different results. The idea of the study is interesting, but the authors do not address the title question. The title is quite general and needs a more detailed analysis before the article is accepted. The author uses only two bacterial species, S. aureus and S. epidermis, so the results cannot be generalized to all bacterial species. I suggest that the author perform a more comprehensive analysis that includes more species or change the title of the article. Another concern is the scripts used: "" ext-link-type="uri" xlink:type="simple">https://github.com/UCF-Li-Lab/Min_AD_project_code" are not documented, so replication of the study is not possible.

7. PLOS authors have the option to publish the peer review history of their article (what does this mean?). If published, this will include your full peer review and any attached files.

Reviewer #1: No

Reviewer #3: **Yes: **Morad M. Mokhtar

---

## [Author Response · Author response to Decision Letter 1]

6 Sep 2023

Reviewer 1 

Comment 1: The article entitled “Are known bacterial strains really present?” seems to be interesting for the research scientific community. In the revised current form authors have shown the good potential in improving this article, but still, it is in lack of structured. This reviewer thinks this article can be improve more by summarizing and pointing few concerns. Start from the abstracts, the provided abstracts could be further improved by incorporating the following elements: (1) The abstract should start with a clear and concise objective statement that highlights the specific research question or problem being addressed. This will help readers understand the purpose of the study right from the beginning. (2) Provide a brief description of the methods or approaches used in the study in one or two statements. This should include information on the data collection process, analysis techniques, and any specific tools used. (3) Instead of just stating that known strains cannot be predicted and are different across tools, provide more specific and quantitative information about the results obtained. This will provide the readers with a clearer understanding of the findings. (4) Discuss the broader implications of the study findings in a single statement. (5) Conclude the abstract by highlighting potential future directions or recommendations based on the study's outcomes. 

Response: Thank you for the great comments! We modified the abstract as you suggested.

Comment 2: (6) In introduction, there is an excessive use of citations/references. In example, only first paragraph of the introduction including 14 references in 7 lines. Please use the citation/references according to the journal guidelines and where appropriate. (7) In introduction where stated “We reran Pathscope2 on these samples …” please clearly mention the validity of findings. Adding potential reasons for the discrepancies and significance of findings. This reviewer suggests to revise thoroughly the introduction section by mentioning the Objective statement with limitations and significances at the end of introduction; 

Response: Thanks for the comments! We rephrased the first paragraph and removed the first right references as you suggested. Note that the first eight references represent different applications and advancement by the next generation sequencing, including ChIP-seq for histone, ChIP-seq for transcription factors, the interaction of transcription factors from ChIP-seq, Hi-C, RNA-seq, metagenomics, strain studies, etc. They are necessary for laymen to understand the broad field of NGS while not all closely related to bacterial strains. We believe that other references are necessary and closely related to bacterial strain studies. We also added some more recent references. 

We reran Pathscope2 because the original study applied Pathscope2 to predict known strains on these samples, and we wanted to see whether the rerun of Pathscope2 would result in a similar set of known strains. Moreover, the pathscope2 and StrainEST tools are more widely used and kept updating after their publication. There are a couple of more recent tools, which are not well tested and cited. Moreover, some of them are not for general bacterial species or only study strains in certain local regions. For these reasons, we chose pathscope2 and StrainEST. The rationale to choose the Chng et al. samples and the two tools are also included in the Methods section. We also added a paragraph to make our objective in the last paragraph of the introduction easier to be understood. The objectives were to show that the available methods have a flawed logic in their design and cannot really prove the existence of known strains. We also pointed out the limitation and significance of our study.

Comment 3: (8) Furthermore, why it is so important to use the data from study of Chng et. al, published in 2016. Please consider using some stats and data comparison from the recent scientific study from last 5 years. 

Response: Thanks for the great comments! In our opinion, the choice of the data does not matter much, as the next generation sequencing technology and their application are already quite mature by 2012, including many analysis tools. We chose the Chng data because we have an interest in S. aureus and S. epidermis. Moreover, there are a large number of their known strains available. We believe our conclusions should hold with other datasets, other bacterial species, and other tools used, because as implied in the paper, all available tools claim the presence of a known strain if a portion or all of its SNPs (k-mers, MLTSs, etc.) occur in reads, which may be wrong and misleading. For instance, the occurrence of all its SNPs in reads could be due to the presence of several unknown strains, whose SNPs include all SNPs of this known strain. 

Comment 4: (9) This reviewer suggests to revise your references very carefully and according to the journal style. Most of the references used in this study are out dated. Please defend your research innovation by comparing the study from the reference list of last five years research. 

Response: Thanks for the very good comments! While it is true that some of the references included in this paper are not within the last five years, they are no less relevant. For instance, the pathscope2 and strainEST are still the most widely cited tools for known strain identification of general bacterial species on the genome scale (i.e., not locally, not for a given set of species). Note that, although some references are published more than five years ago, our study is still novel, as it shows novel insights about the limitations of the current known strain identification tools that scientists do not realize. In brief, available tools claim the occurrence of a known strain based on the occurrence of its SNPs or others in reads, which does not guarantee the occurrence of this known strain at all. We probably need to directly apply the novel strain prediction tools and experimentally validate the existing of these strains by long read sequencing. 

Comment 5: (10) Please revise the manuscript carefully, and avoid the grammar mistakes. I suggest you to invite a native speaker to proofread this article before further submission, or use any official English language proofread services.

Response: Point (10): Thank you for the suggestion. We carefully modified the manuscript and have two native speakers proofread the manuscript. 

Reviewer 2: 

Comment 1: The case study described aimed to answer the title question by revisiting the known strains of S. aureus and S. epidermidis in atopic dermatitis shotgun metagenomic samples. The findings of the study revealed that even using the same analysis tools, the previously reported known strains could not be predicted consistently in the same samples. Additionally, different tools gave different results. The idea of the study is interesting, but the authors do not address the title question. The title is quite general and needs a more detailed analysis before the article is accepted. The author uses only two bacterial species, S. aureus and S. epidermis, so the results cannot be generalized to all bacterial species. I suggest that the author perform a more comprehensive analysis that includes more species or change the title of the article. 

Response: Thank you for the great comments! We modified the title to emphasize that this is a case study and its conclusion may need to further comprehensively validate with more species and datasets. As we commented above, the conclusions made with this datasets and two species should be valid for other species and datasets as well, especially for those fast evolving bacterial species.

Comment 2: Another concern is the scripts used: "https://github.com/UCF-Li-Lab/Min_AD_project_code" are not documented, so replication of the study is not possible.

Response: Thanks for the helpful comment. We updated the readme file and the link to help readers to replicate the analysis in this study.

---

## [Decision Letter · Decision Letter 2]

10 Sep 2023

Are the predicted known bacterial strains in a sample really present? A case study

PONE-D-22-31800R2

Dear Dr. Li,

We’re pleased to inform you that your manuscript has been judged scientifically suitable for publication and will be formally accepted for publication once it meets all outstanding technical requirements.

Kind regards,

Achraf El Allali, PhD

Academic Editor

PLOS ONE

Additional Editor Comments (optional):

Reviewers' comments:

Reviewer's Responses to Questions

**Comments to the Author**

1. If the authors have adequately addressed your comments raised in a previous round of review and you feel that this manuscript is now acceptable for publication, you may indicate that here to bypass the “Comments to the Author” section, enter your conflict of interest statement in the “Confidential to Editor” section, and submit your "Accept" recommendation.

Reviewer #1: All comments have been addressed

Reviewer #3: All comments have been addressed

2. Is the manuscript technically sound, and do the data support the conclusions?

Reviewer #1: Partly

Reviewer #3: (No Response)

3. Has the statistical analysis been performed appropriately and rigorously? 

Reviewer #1: Yes

Reviewer #3: (No Response)

4. Have the authors made all data underlying the findings in their manuscript fully available?

Reviewer #1: Yes

Reviewer #3: (No Response)

5. Is the manuscript presented in an intelligible fashion and written in standard English?

Reviewer #1: No

Reviewer #3: (No Response)

6. Review Comments to the Author

Reviewer #1: The authors has addressed all the concerned issues. But article has Grammatical mistakes. I suggest you to revise it again before the final publication.

Good luck!

Reviewer #3: (No Response)

7. PLOS authors have the option to publish the peer review history of their article (what does this mean?). If published, this will include your full peer review and any attached files.

Reviewer #1: No

Reviewer #3: **Yes: **Morad M. Mokhtar

---

## [Editor Report · Acceptance letter]

6 Oct 2023

PONE-D-22-31800R2 

Are the predicted known bacterial strains in a sample really present? A case study 

Dear Dr. Li:

I'm pleased to inform you that your manuscript has been deemed suitable for publication in PLOS ONE. Congratulations! Your manuscript is now with our production department. 

Kind regards, 

on behalf of

Dr. Achraf El Allali 

Academic Editor

PLOS ONE